# Experimental Study of a Tilt Single Slope Solar Still Integrated with Aluminum Condensate Plate

**Naseer T. Alwan** [1,2,*], **Milia H. Majeed** [1], **Sergey E. Shcheklein** [1], **Obed M. Ali** [3]
**and Seepana PraveenKumar** [1,*]

1   Department of Nuclear and Renewable Energy, Ural Federal University Named after the First President of Russia B. N. Yeltsin, 19 Mira St., 620002 Yekaterinburg, Russia; milia.hameed88@gmail.com (M.H.M.); s.e.shcheklein@urfu.ru (S.E.S.)
2   Technical Engineering College of Kirkuk, Northern Technical University, Kirkuk 36001, Iraq
3   Renewable Energy Research Unit, Northern Technical University, Kirkuk 36001, Iraq; obedmajeed@gmail.com
*   Correspondence: nasir.al-kharbavi@urfu.ru (N.T.A.); ambatipraveen859@gmail.com (S.P.)

**Abstract:** The low freshwater productivity of a conventional solar still is considered a challenge for researchers due to the high temperature of the glass cover or basin water depth. In current work, a newly designed solar still was suggested according to the climatic conditions of Yekaterinburg/Russia, which included an enhanced condensation and evaporation process by spraying a thin water film on a hot absorber plate and then passing the generated water vapor by free convection over the aluminum plate (low temperature). The distillation system under study was tested during July 2020 and 29 July was chosen as a typical day from 08:00 a.m. to 8:00 p.m. The results showed that the largest amount of water vapor condenses on the aluminum plate (about 46%), and the rest condenses on the glass cover. This means that the aluminum plate effectively improved productivity due to the flow of humid air naturally (free convection) on the aluminum plate (its surface temperature was lower than that of the glass cover). The cost analytical calculations showed that the cost of producing one liter of distilled water from the suggested solar still was 0.063$.

**Keywords:** a tilt single slope solar still; aluminum condensate plate; absorbent plate





## 1. Introduction

The percentage of fresh water is about 2.5% of the total water that covers about 70% of the earth's surface. Seawater contains high levels of salinity ranging from 3500 to 4500 ppm, and according to reports of the World Health Organization (WHO), the acceptable proportions of dissolved salt in drinking water are generally 500 ppm and, in some special cases, 1000 ppm [1]. This makes seawater unsuitable for direct human consumption and for agriculture and industrial utilization without a desalination process [2]. The average daily production of fresh water from the desalination process worldwide is about $23 \times 10^6$ m$^3$ [3]. However, fresh water production process requires significant fossil fuel consumption. Studies have shown that approximately 130 million tons of oil (annually) is used to produce 13 million cubic meters of potable water [4].

According to 21st century data released by the Renewable Energy Policy Network in 2016, the dominant energy source to meet world demand is fossil fuel, which constitutes about 78.3%, while renewables constitute 19.2% of total energy, and the remaining 2.5% represents nuclear fuel [5].

Therefore, numerous strategies need to be planned to use renewable energy sources (solar energy) as an alternative energy source. There are generally two ways of using solar energy; (i) direct use of thermal energy where there is no need for energy conversion, i.e., electrical energy; (ii) using photovoltaics technology by absorbing solar energy through PV solar panels.

Solar water distillation technology is considered as one of the modern trends to provide potable freshwater from alternative energy, particularly in rural and remote areas [6]. The distillation through solar energy is a process that produces potable water from brine water, untreated water, polluted water, or brackish water by consuming free energy from the sun without any fuel consumption. The distillation process includes two methods; natural evaporation and natural condensation, which ultimately remove impurities such as salts and bacteria, giving potable water in addition to diminishing the environmental effects [7]. The desalination technologies in solar still emulate the natural hydrological cycle process of evaporation and condensation.

Generally, the performance of solar stills is affected by three parameters: meteorological (climate), design, and operation parameters [8]. The intensity of solar radiation is the main effective parameter that defines the overall efficiency, however, the ambient temperature, wind speed, etc., can also affect the performance of the solar plant. Among the most important factors that can affect the solar still's performance are the design parameters that include; thermo-physical properties of the materials used in solar stills, the slope angle of the transparent cover of the solar stills, and the dimensions and shape of the solar stills, etc. [9]. These parameters can be controlled and modified depending upon the climate conditions and type of application. On the other hand, the operating parameters include; basin water depth [10,11], still orientation, water salinity, and initial basin water temperature, etc. [12].

The main indicator for the solar still's performance efficiency in the production of fresh water is the surface area available to enhance productivity. Many attempts have been carried out to enhance the productivity of conventional basin-type solar stills that have relatively low productivity. The conducted studies focused on modifications including the primary factors affecting the solar still performance: installation of the condensing unit, energy absorption and storage materials, heating units, and rotating shafts or cylinders inside the solar still. The rate of vapor condensation and basin water evaporation can be increased by elevating the temperature difference among the basin water and other surfaces inside the solar still; as a result, the freshwater yield will enhance. Naseer T. Alwan et al. [13] suggested a new integration between the single-slope solar still and the photo-electric diffusion–absorption refrigerator (DAR) to increase the freshwater productivity of the conventional solar stills. The results revealed an increase in the daytime freshwater productivity of about 251% after modifications and 470% during the night compared to the unmodified solar still. Husham [14] carried out an investigation on the effect of integrating a built-in passive condenser into a conventional solar still on the daily cumulative distillate water. It was found that solar still with a built-in condenser gives about 16.7% higher cumulative distillate water as compared to the conventional solar still. There are many methods that have been used to increase the amount of solar radiation reaching the surface of the solar still, such as increasing the amount of solar energy absorbed, for example, adding coal to the basin water in the solar still increases the energy required for evaporation as a result of absorbing more solar radiation [15]. The study [16] used paraffin wax as a latent heat material in a solar distiller; the study included an experimental investigation about the effects of incorporating Phase change materials (PCM) cells with a solar still basin and compared its performance and daily productivity with that of a conventional solar still without PCM cells. The results showed that the productivity of the proposed device with PCM was improved by 32% compared to that of the conventional type. An experimental and theoretical study integrated an evacuated-tube solar heater with a single-slope solar distiller. In the climate conditions of Turkey, the results showed that productivity was increased, and a mathematical model was provided, which had good agreement with experimental data [17].

To increase the surface area of the evaporation part and decrease the thickness of saltwater of the solar distiller, in previous studies a rotating shaft or cylinder inside the solar distiller was added. A thin water film formed over the hot rotating surface and evaporated rapidly, and was constantly being generated compared to the thickness of the

layer of saltwater in a traditional solar distiller. Essa et al. [18] enhanced water distiller productivity by changing a traditional solar distiller to a modified solar distiller integrated with rotational discs (flat discs and corrugated discs with and without wick material) to increase the surface of the evaporation area and decrease the thickness of saline water. Eight rotational speeds were tested (0.02, 0.05, 0.1, 1, 0.5, 2, 3, and 4 rpm). The results showed that the daily productivity of the modified model was greater than that of the traditional solar distiller, and the best performance was with 0.05 and 0.1 rpm with and without wick material. The improvement rate was 124% compared to the traditional solar distiller with rotational corrugated discs having a wick. The maximum thermal efficiency was found to be 54 and 50% with a corrugated and flat disc and wick at 0.05 rpm, respectively. A study [19] was carried out to investigate the effect of a hollow cylinder rotation inside a solar still integrated with an external solar collector. The study was carried out in two stages; in the first stage, three rotational speeds were tested (0.5, 1, and 3 rpm) for the modified solar still [20,21]. The second stage involved the integration of the solar water collector with the modified solar water collector [22]. The results showed that in the first stage, the maximum productivity was obtained at 0.5 rpm, and the second stage improved the productivity by 292%.

Various studies have been conducted to improve the freshwater productivity of solar stills by controlling design and operation parameters. Therefore, the present study aims to present a new and simple design for a solar distiller to convert saltwater into potable water (in areas where saltwater is available) or extract moisture from moist air (for areas with relatively high humidity) by passing a thin water film over a hot absorber plate to accelerate the evaporation process and then pass the moist air over an aluminum plate (low temperature) by free convection. The suggested solar still is considered a pioneering attempt to provide freshwater with a simple design and low cost, especially in remote and rural areas.

## 2. Materials and Methods

The condensation process is the deposition of water vapor from the ambient air on exposed surfaces that are cooler than the surrounding air, such as leaves of plants, trees, glass panels, plates, etc. The precipitate is in the form of droplets of liquid water called dew, which usually forms during the night or shadow times when the air is still, or the wind is light. The formation of dew is because most of the exposed objects radiate more heat from the ambient air, making it colder than it is. The temperature at which dew forms on these bodies is called the dew point, which is the temperature at which the air surrounding objects reaches the saturation point.

According to this principle, a proposed solar still system was designed to extract fresh water from the moist air by passing it over a cooler surface (finned aluminum plate) that is naturally cooled by the surrounding air currents. The saltwater is sprayed from the top of the solar still onto the absorption plate by small nozzles. A micro-water pump was used to circulate the water between the solar still and the insulated water tank (40 cm long, 20 cm wide, and 10 cm high) with a very low flow rate of 0.066 L/min, which was equipped with power from a 20 WPV panel. The absorbent plate was heated by solar radiation, when a water film, as droplets, fell on this hot plate and evaporated, a part of the water vapor stuck and condensed on the inner surface of glass cover of solar still, and the rest was carried by the light air stream upwards (due to the buoyancy force) to pass it over the cold aluminum plate and thus condense water vapor on its surface, as shown in Figure 1.

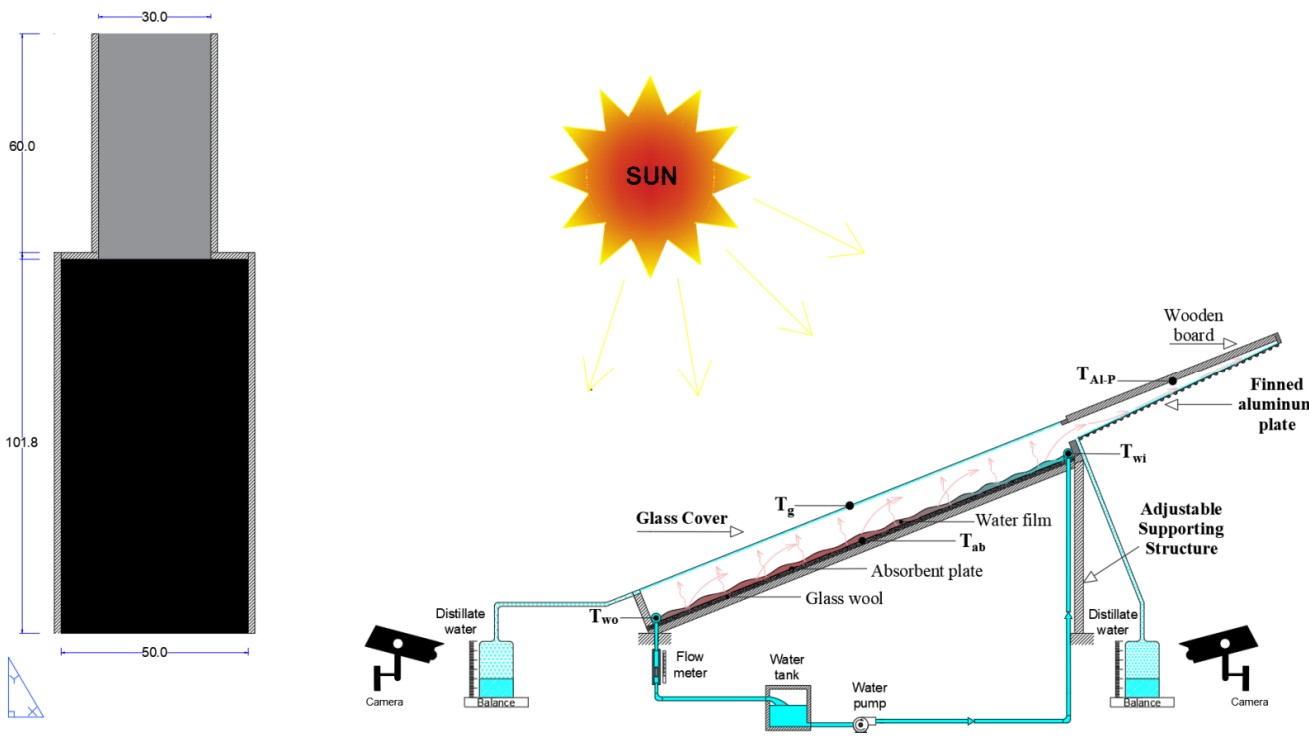

**Figure 1.** A schematic diagram of the suggested solar still.

### 2.1. Experimental Setup

Figure 2 shows a photographic view of the tilt solar still. The distillation system had a dimension of 100 cm length, 50 cm width, and 10 cm depth, and the solar water still was tilted at 25° [23]. The proposed solar still mainly consisted of a wooden case, a glass cover, absorber plate, two aluminum channels to collect distilled water, nozzles with a feed water pipeline, micro water pump, flow meter, and water tank. The wooden case consisted of a frame with 100 cm length, 50 cm width, 10-cm height, and 1.8 cm thickness, as well as a Medium density fiberboard (MDF) board of 100 cm length, 50 cm width, and 1.8 cm thickness on the backside of solar still. The glass cover was 100 cm in length, 50 cm width, and 0.4 cm thickness. The bottom solar water distiller was covered by a stainless-steel absorption plate of 100 cm length, 50 cm width, and 0.1 thickness, and it was coated with matte black to increase its absorption capacity to sunlight. Glass wool of 1 cm thickness was fitted between the MDF board and the absorber plate to reduce the heat loss from the absorber plate.

At the top of the solar still, a rectangular duct was installed with dimensions of 60 cm length and 30 cm width. The bottom surface of the channel was a sheet of aluminum finned with 0.1 cm thickness, while the top surface of the duct was covered by an MDF board to prevent sunlight from reaching the surface of the aluminum plate. The aluminum plate tilted was 30 degrees from the horizon to force moist air to contact its surface.

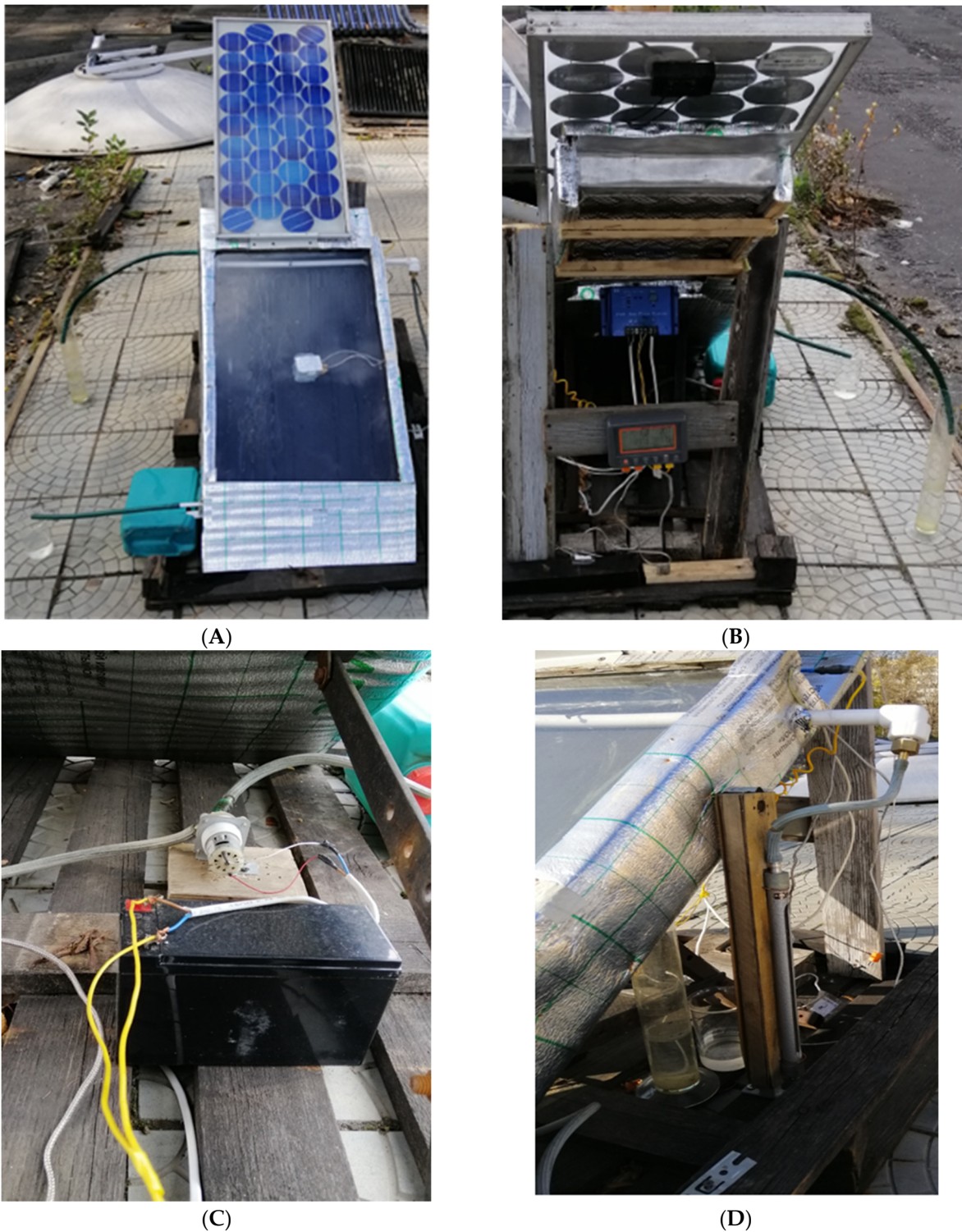

**Figure 2.** A photograph of the suggested solar still. (**A**) Front side, (**B**) back side, (**C**) micro pump and battery, (**D**) flow meter.

### 2.2. Experimental Method and Uncertainly Analysis

The experimental data was collected between 08:00 a.m. to 8:00 p.m. during July 2020 and 29 July was chosen as a typical day. The test rig was established in the Ural Federal University, Ekaterinburg, Russia (56.8431° N, 60.6454° E). For receiving maximum solar radiation, the solar still was oriented to the south direction. The data was recorded hourly, such as the absorber plate temperatures, glass cover, the inlet water to the solar still, the

outer water from the solar still, aluminum plate temperature, ambient air, solar radiation intensity, and freshwater productivity. An SD data logger with four channels module 88598 (AZ Instrument Corp, Taiwan, China) was used to collect the temperature at certain points of solar still, while the ambient air temperature was measured using a thermometer. A solar power meter device type TM207 supplied by Instruments zone, India was used to measure the solar radiation intensity.

In an experimental study, uncertainty analysis is an important and necessary step to give credibility and confidence to the results. Therefore, at the first step, one must know the accuracy values of each measuring instrument used in the experimental process [16]. Table 1 lists the measuring equipment accuracy value and the measurement error range, which have been calculated using the following equations:

$$S = \sqrt{\frac{\sum_{i=1}^{n} \left(X_i - \acute{X}\right)^2}{n-1}} \tag{1}$$

$$S.E = \frac{S}{\sqrt{n}} \tag{2}$$

$$E\ \% = \frac{S.E}{\acute{X}} \times 100 \tag{3}$$

$$X' = \sqrt{\frac{\sum_{i=1}^{n} X_i}{n}} \tag{4}$$

**Table 1.** Experimental device uncertainty analysis results.

| Equipment | The Accuracy | The Range | The Error Ratio % | Units |
|---|---|---|---|---|
| Data logger | 1 °C | −200–1370 | 0.3% | °C |
| Thermocouple | 0.1 °C | −100–200 | 0.3% | °C |
| Mercurial thermometer | 1 °C | 0–100 | 0.5% | °C |
| Solar power meter | 0.1% | 0–2000 | 0.1% | W/m$^2$ |

From the above equations, the standard deviations (*S*) and standard error (*SE*) have been calculated based on the measured value (*X$_i$*) and their mean value (*X*), which in turn is used to calculate the error (*E*) for the specified number of times measured (*n*).

## 3. Results and Discussion

The solar radiation intensity, ambient air temperature, and wind speed are considered as the most important parameters affecting the performance of a solar still. Figure 3 shows the hourly solar radiation intensity and the ambient air temperature between 8:00 a.m. and 8:00 p.m. for the typical day of 29 July 2020. After 8:00 a.m., there was a gradual increase in the intensity of solar radiation, and it reached its highest value at 1:00 p.m. of around 989.9 W/m$^2$. It gradually decreases after 1 p.m. until the end of sunrise at 8:00 p.m. In the morning after sunrise, sunlight transmitted thermal energy to areas on the surface of the earth and the surrounding air [21], so the highest ambient air temperature was recorded at 15:00 p.m., of around 30.5 °C. Wind speeds were moderate during the day of the test, with values ranging from 2.1 to 4.1 m/s.

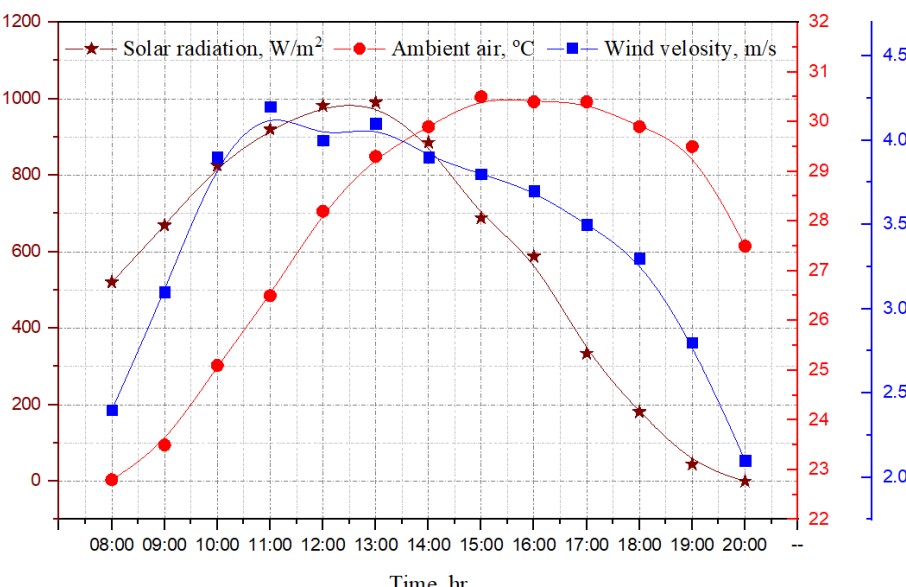

**Figure 3.** The hourly change in solar radiation intensity, ambient air temperature, and wind speed for a typical day, 29 July 2020.

Figure 4 illustrates the hourly variation of the absorber plate temperatures; inlet and outlet water temperatures, glass cover temperature, aluminum plate temperature, and ambient air temperature for suggested solar still. From this figure, we noted that the temperatures of the absorber plate, the water at inlet and water at outlet, and the glass cover were low in the early hours of the test (from 8:00 a.m. to 10:00 a.m.) due to the time required to raise the temperature of the absorbent plate by solar radiation, which was insufficient because of its thickness; in addition, the circulating water takes time to raise its temperature by passing it to the absorber plate to gain thermal energy from the solar radiation and the absorbent plate, then to the water tank to be sprayed again through the nozzles on the absorbent plate surface. In comparison, the glass cover needs less time to raise its temperature because the heat capacity of glass is lower than that of water and the absorber plate. Thereafter, temperatures at different points of the solar still continued to increase in conjunction with the increases of solar radiation intensity and recorded the highest values around at 1 p.m. about 58.1 °C for absorbed plate, 54.8 °C for water at inlet solar still, 57.37 °C for water at outlet solar still, and 50.24 °C for glass cover. From this figure, it was observed that the temperature of the aluminum plate was directly affected by the ambient air temperature and wind speed. The aluminum plate recorded the highest temperature around 3:00 p.m. of about 33.5 °C when the ambient air temperature recorded the highest value about 30.5 °C.

Figure 5 shows the hourly freshwater yield from both the aluminum plate and the glass cover for the typical day 29 July 2020. From this figure, we observed that the aluminum plate condensed a larger amount of the water vapor (about 46%) and the rest condensed on the glass cover. Because the evaporator temperature was lower, and that depended on the weather conditions. The highest yields from the aluminum plate and glass cover were recorded at midday, at approximately 140 and 100 mL/m²h, respectively, when the solar radiation intensity was at the highest value of approximately 989.9 W/m² (higher water evaporation rate), the ambient air temperature was 29.3 °C, and the wind speed was 4.1 m/s.

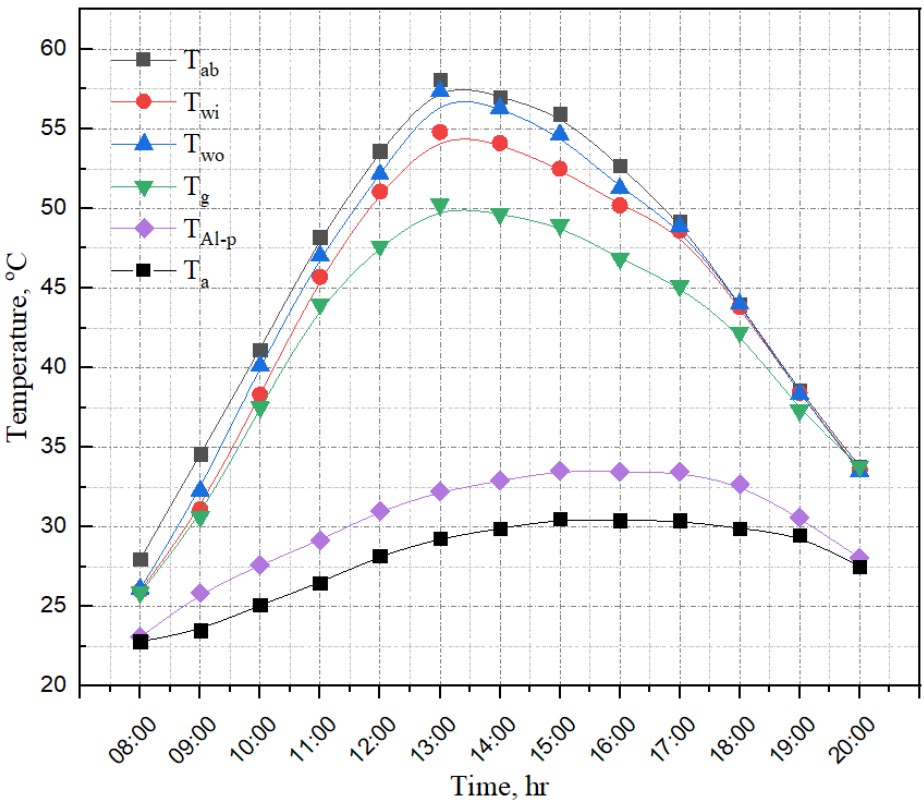

**Figure 4.** Hourly change temperatures in the absorber plate; inlet and outlet water, glass cover, aluminum plate, and ambient air for a typical day, 29 July 2020.

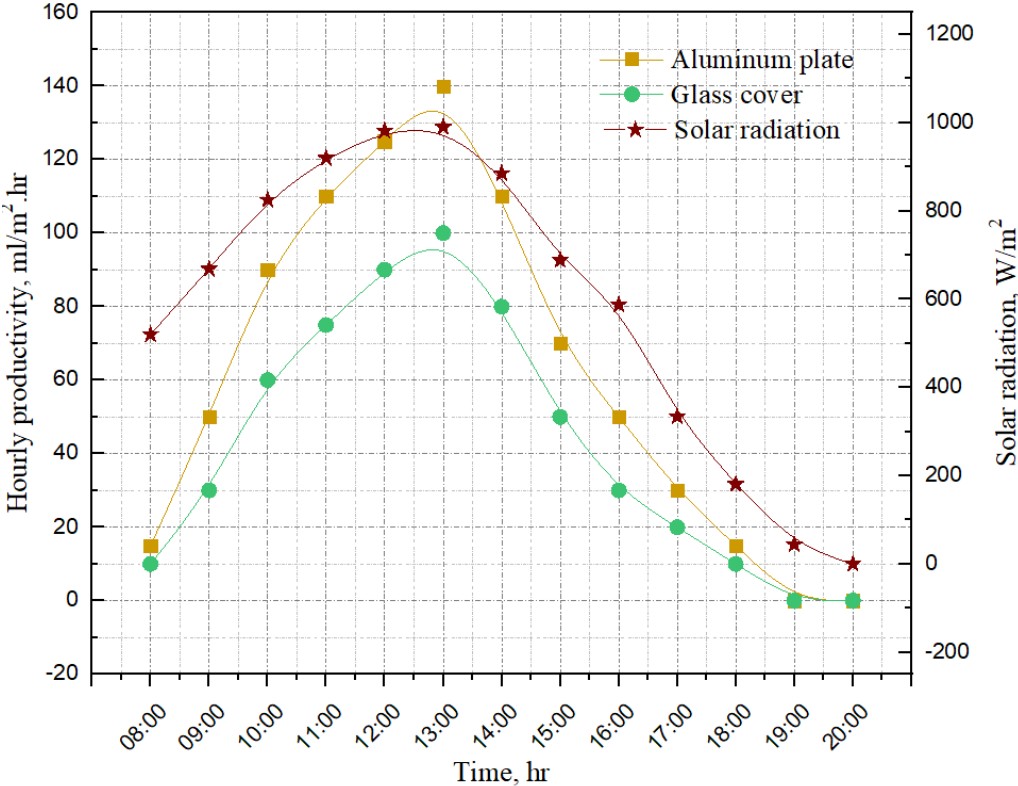

**Figure 5.** Hourly variation in the solar radiation with distillate yield from aluminum plate and glass cover for a typical day, 29 July 2020.

Figure 6 shows a cumulative freshwater from an aluminum plate, glass cover, and total during 12 h from 8:00 a.m. to 8:00 p.m. for a perfect day, 29 July of 2020. From this Figure, it was observed that the cumulative productivity from the aluminum plate was higher compared with the glass cover, which recorded about 805 mL/m², while from the glass cover it was 555 mL/m². This means that the aluminum plate had an influential effect in improving productivity due to the flow of humid air naturally (free convection) on the aluminum plate (its surface temperature was lower than that of the glass cover).

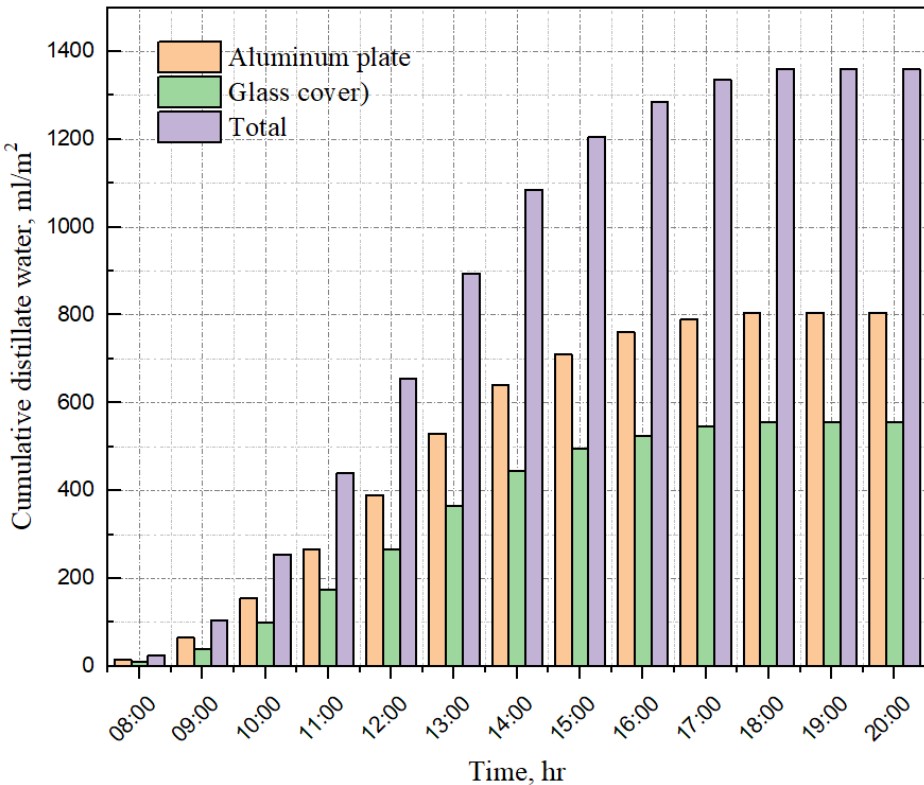

**Figure 6.** Cumulative freshwater production from aluminum plate and glass cover for a typical day, 29 July 2020.

Figure 7 shows the hourly efficiency of the suggested solar still during 8 h from 9:00 a.m. to 4:00 p.m., which represents active working hours. The hourly efficiency of the suggested solar still was calculated by multiplying the hourly cumulative distillate water output by the average latent heat, then the results were divided by the hourly solar radiation over the whole area A (0.5 m²) and period (3600 s) and power consumption of the water pump (6W). From this Figure, it was noticed that the system efficiency was generally low (the highest value was about 9.3 at 2:00 p.m.), due to the lower production rate relative to the solar energy absorbed by the absorber.

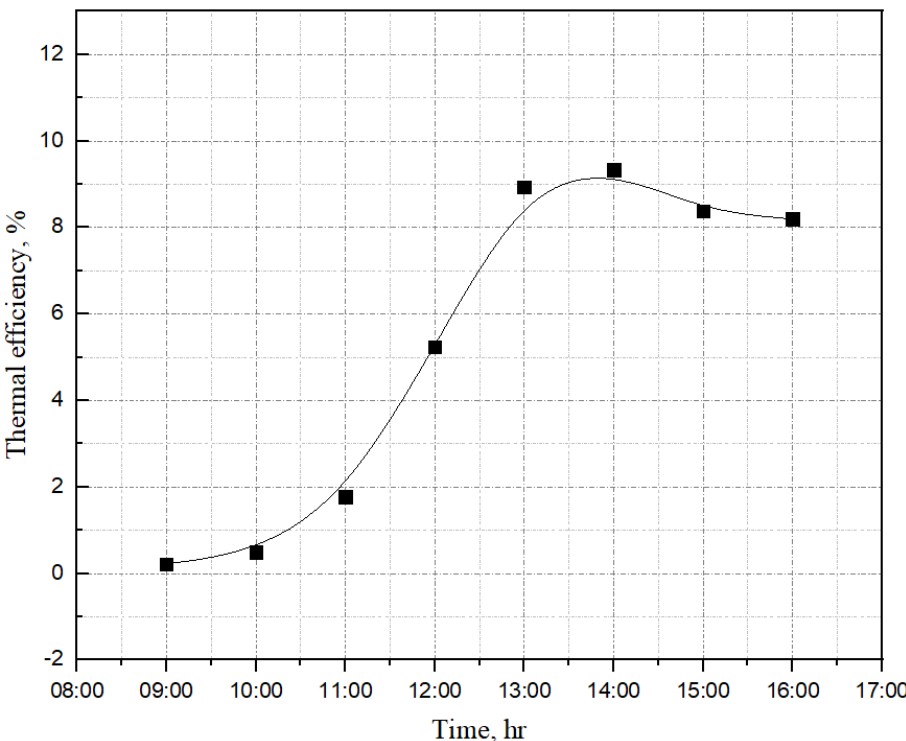

**Figure 7.** Hourly thermal efficiency of solar still for a typical day, 29 July 2020.

## 4. Analysis of Production Cost

Improvement of the freshwater yield from solar stills is not only the primary goal, but also the cost of production must be taken into consideration. Different studies have carried out detailed economic analyses of the important factors involved in the cost of producing one liter of freshwater [23], such as capital cost (CS), the capital recovery factor (CRF), the sinking fund factor (SFF), the first annual cost (FAC), the annual salvage value (ASV), the annual cost (AC), the annual maintenance cost (AMC), and yearly cost per one liter (YCL). The capital recovery factor, the sinking fund factor, and the first annual cost (FAC) can be calculated as follows;

$$CRF = \frac{i * (1 + i)^n}{\left[ (1 + i)^n - 1 \right]} \tag{5}$$

$$SFF = \frac{i}{\left[ (1 + i)^n - 1 \right]} \tag{6}$$

$$FAC = CRF * CS \tag{7}$$

The salvage value of water distillate was considered 0.2 of the Capital cost (CS) [24], so, the annual salvage value (ASV) calculated as:

$$ASV = S * SFF \tag{8}$$

We also considered that the annual maintenance cost (AMC) was 0.15 of the first annual cost (*FAC*), so, the annual cost (AC) was calculated as [25]:

$$AC = FAC + AMC - ASV \tag{9}$$

The yearly cost per liter (*YCPL*) is as follows:

$$YCL = \frac{AC}{L} \tag{10}$$

where *L* is the annual production of fresh water from the suggested solar distiller.

Table 2 includes the details of the capital cost of the proposed solar still in the Russian local market. It was noted from this table that the total cost of the suggested solar still was 80.5$. Table 3 included an analysis of the annual production cost of one liter of distilled water. The cost analytical calculations showed that the annual cost of producing one liter of distilled water from the suggested solar still was 0.063$. To find out the feasibility of the suggested solar still, a comparison was with the previous studies (the annual cost of producing one liter of distilled water), as shown in Table 4. From this Table, a good agreement was observed between previous studies and the suggested solar still.

**Table 2.** Manufacturing and installation capital cost of solar stills, $.

| Type Material | Quality | SSS ($) |
|---|---|---|
| MDF Wooden Board 1.8 cm thickness | 2 m$^2$ | 14 |
| Glass cover 0.4 cm thickness | 0.5 m$^2$ | 1.5 |
| Galvanized stainless sheet, 0.1 cm | 0.5 m$^2$ | 2 |
| Alminum plate | 0.2 m$^2$ | 5 |
| PV panel and accessories | 1 piece | 50 |
| Micro water pump | 1 piece | 2 |
| Spray paint heat-resistant | 2 pieces | 3 |
| Heat-resistant silicone glue | 2 pieces | 3 |
| Total cost | - | 80.5 |

**Table 3.** Unit costs analysis for water produced in dollars.

| The Item | Suggested Solar Still | The Item | Suggested Solar Still |
|---|---|---|---|
| Life expectancy for solar still, n | 10 | The value of salvage (S), $ | 16.1 |
| Interest rate per year i,% | 12 | Annual salvage value (ASV), $ | 0.916 |
| manufacturing and installation capital cost (CS), $ | 80.5 | Annual maintenance cost (AMC), $ | 2.136 |
| Capital recovery factor (CRF), | 0.1769 | Annual cost (AC), $ | 15.459 |
| Sinking fund factor (SFF) | 0.0569 | Yearly yield from the solar still system (180 days), L | 244.8 |
| First annual cost (FAC), $ | 14.240 | Yearly cost per liter ($YCL$), $ | 0.063 |

**Table 4.** The production cost comparison with other studies.

| Studies | Solar Water Distiller Type | The Location | Daily Productivity, L/m$^2$ | Productivity Cost, l/$ |
|---|---|---|---|---|
| [23] | The single slope solar distiller | Egypt | 8.39 | 0.035 |
| [26] | The single slope solar distiller | India | 1.91 | 0.14 |
| [27] | The single slope solar distiller | Pakistan | 3.25 | 0.063 |
| [16] | The single slope solar distiller | Iraq | 2.35 | 0.035 |
| [19] | The single slope solar distiller integrated with solar water collector | Russia | 5.5 | 0.0477 |
| Current study | A tilt single slope solar still integrated with aluminum condensate plate (ACP) | Russia | 1.36 | 0.063 |

## 5. Conclusions

The current work provides experimental results of integrating a finned aluminum plate with a tilt single slope solar still, and concluded the following:

1. The suggested solar still is considered a pioneering attempt to produce freshwater water with a simple design and low cost, especially in remote and rural areas, by converting salt water into potable water (in areas where salt water is available) or extracting moisture from moist air (for areas with relatively high humidity).
2. The largest amount of water vapor condenses on the aluminum plate (about 46%), and the rest condenses on the glass cover. This means that the aluminum plate gave a significant effect in improving productivity.
3. The system efficiency was generally low (the highest value was about 9.3 at 2:00 p.m.) due to the lower production rate relative to the solar energy absorbed by the absorber.
4. In general, the estimated production cost of one liter of freshwater form suggested solar still is 0.063$. When compared with other studies, agreement was good with it in terms of the cost of production per liter of freshwater.
5. In the future, it is proposed to implement this new form of solar still by adding the filament to the absorbent plate, and the suggested solar still area should be greater than 1 m$^2$.

**Author Contributions:** Conceptualization, N.T.A., M.H.M., S.P.; methodology, N.T.A., S.P.; software, N.T.A.; validation, N.T.A., O.M.A., S.E.S.; formal analysis, M.H.M., S.P.; investigation, S.P.; resources, S.P., N.T.A.; data curation, N.T.A.; writing—M.H.M., N.T.A., O.M.A., S.P.; writing—N.T.A., M.H.M., S.P.; visualization, N.T.A., S.P.; supervision, S.E.S.; project administration, N.T.A., S.P.; funding acquisition, S.P. All authors have read and agreed to the published version of the manuscript.

**Funding:** This research received no external funding.

**Conflicts of Interest:** The authors declare no conflict of interest.

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
