# Peer review of "Experimental Study of a Tilt Single Slope Solar Still Integrated with Aluminum Condensate Plate"

_inventions, doi:10.3390/inventions6040077_

Round 1

Reviewer 1 Report

The manuscript "Experimental study of a tilt single slope solar still integrated with aluminum condensate plate" provides an interesting study about integrating an aluminum plate into a solar-driven salt-water distillation setup to improve distillation rates and efficiency. For the most part, the methods and experiments are explained well and the manuscript is well organized. However, due to two maim problems with the manuscript it can not be recommended for publication at this time:

1) There are many grammatical errors throughout the manuscript that make it difficult to read and review. 

2) The important conclusion is made that aluminum plate improved the productivity of the system. This may very well be true, but to prove this a control experiment should be run without the aluminum plate to show and compare the total rate of water distallation for with and without plate conditions.

There are also many minor typos and minor errors throughout the manuscript, some of which are:

  • Abstract - the cost analytical calculations showed that the annual cost of producing one liter of distilled water from the suggested solar still was $0.063. (This needs to be written more clearly)
  • [35] – Twenty should not be capitalized.
  • [138] – “-20” should be “20-“
  • [139] – W instead of Watts.
  • [199] – m2 instead of m2.
  • [202] – units are missing on 30.5.
  • Regarding Figure 4, it would be helpful to put the positions of the thermocouples in one of the diagrams (maybe Figure 2) and the symbols in the legend can be written in brackets in the caption of Figure 4, just after the location is mentioned.
  • [213] The heat capacity of glass is not larger than that of water.
  • [235-236] – This statement has not been proven. A control experiment should be run with just the absorber (without the aluminum plate) to show that the total amount of water collected is greater with the aluminum plate.
  • [228] – in the units: should be “” instead of “.”
  • [242] – here the ^ symbol is used which is not consistent.

Author Response

All the suggestions and questions were answere according to reviewers request

Reviewer 2 Report

Brief Description of the Work

The authors proposed, and studied experimentally, a solar still system designed to extract fresh water from the moist air by passing it over a cooler finned aluminium plate that is naturally cooled by the surrounding air currents. More specifically, the absorbent plate is heated by solar radiation, when a water film as droplets falls on this hot plate, it evaporates, a part of the water vapour condenses on the inner surface of glass cover of solar still, and the rest is carried by the light air stream to upwards to pass it over the cold aluminium plate and thus condense water vapour on its surface.

Main Obtained Results

- The aluminium plate gave an effective effect in improving productivity as the largest amount of water vapour condenses on the aluminium plate about 46%, and the rest condenses on the glass cover.

- The estimated annual cost for the production of one liter of freshwater from the suggested solar still is 0.063$.

- The system efficiency is generally low due to the lower production rate relative to the solar energy absorbed by the absorber.

General Considerations (GC)

GC1) The manuscript is interesting and well written; I enjoyed reading it.

GC2) Please check the English of the manuscript; some typos have been detected.

GC3) The authors concluded that the performance of the system efficiency is low due to the lower production rate relative to the solar energy absorbed. This is certainly correct, however, as rightly mentioned in the manuscript. the performance of the solar still is also affected by many other factors, such as the ambient conditions, the operating conditions, and the design conditions. The main purpose of the questions below is to get more information, and further clarifications, about the impact of these factors on their solar still system.

Questions/Suggestions (Q/S)

Q/S1) The distillation productivity of solar stills is significantly influenced by ambient conditions. The authors analysed the hourly change in solar radiation intensity, ambient air temperature and wind speed - for typical day 29/07/2020 (ref. to Figure 3.). However, other two important factor may influence the productivity: dust and cloud cover. Have the authors collected experimental data concerning these two ambient conditions? In case of positive answer, may the authors provide with some graphics about?

Q/S2) As climatic conditions are beyond our control, the performance of a solar still design output can be further improved via operational and design conditions.

The authors did not sufficiently mention the impact of the depth of water, various dyes, salt concentration, and inlet temperature of water on the performance of their tilt single slope solar still. The authors are therefore asked to add some supplementary information about the impact of these operating condition on the performance of their solar still system.

Q/S3) The authors did not sufficiently describe the impact of the slope of the cover, storing materials, insulation, and gap distance on their design solar still. Even in this case, the authors are asked to add some supplementary comments about the impact of these design conditions on their solar still system.

Q/S4) As mentioned above, one of the main conclusions of the authors is that the efficiency of their system is generally low due to the lower production rate relative to the solar energy absorbed by the absorber. However, it is well-established that the application of both internal and external reflectors increases the amount of absorbed solar radiation on the basin liner. The potential output of a basin type still can increase to almost 70–100%. May the authors add some information concerning the impact of their internal and external reflectors on the potential output of the system?

Q/S5) As said, the daily productivity is enhanced by the application of the reflectors. The efficiency is increased from 48% to more than 70% when coupled with outside condenser. The storage systems may be used for enhancing the distillate production of solar stills. Have the authors analysed the possibility of incorporating the outside condenser (coupled with the reflectors) and the storage system in their design solar still. In case of positive answer, may the authors add some information about?

Q/S6) Is the designed solar system still equipped with the sun tracking system? Indeed, it is now established that the sun tracking system is more effective than fixed system and it is capable of enhancing the productivity of the still.

Q/S7) The rate of distillation of a typical solar still design is usually very slow: 6 litres of water per sunny day. Reading the manuscript, I did not find the quantity of distilled water for sunny day produced by the proposed solar still system. In a nutshell, how many liters are produced per day for sunny day? Can we affirm that even the solar still system proposed by the authors is not suitable for larger consumptive needs?

Q/S8) As known, one of the main disadvantages of the current solar still systems is that the materials required for the distiller may be difficult to obtain in some areas. May the authors add some comment about this aspect in relation to their design solar still?

Q/S9) Due to the existence of different methods of cost estimation, it is not possible to determine a universal, comparable price per technology. In general, we may affirm that the annual cost per liter of distilled water obtained from the basin type solar still ranges from 0.035$/liter to 0.074$/liter. Table 3 in the manuscript shows an analysis of the annual production cost of one liter of distilled water. The final figure is 0.063 $/liter corresponding, however, to twice the minimum cost (~ 1,8 times) of the cost of the distilled water obtained from a typical basin type solar. We may object that the cost of distilled water from the authors' solar still system is too high. The authors are asked to dispel this possible objection.

Conclusions

As said, the work is very interesting and stimulating and this explains the number of my questions. In my opinion it deserves to be published. However, as the authors know, the questions raised above concern the topics currently dealt with by research groups working in this field. I encourage the authors to take into account the suggestions expressed above; in my opinion, this will increase the interest of many readers.

Author Response

All the suggestions & questions were considered and  answered by the authors, Thank you!

Reviewer 3 Report

Review of an article entitled Experimental study of a tilt single slope solar still integrated with aluminium condensate plate

  1. Why is the article/device about Yekaterinburg / Russia? This, in theory, narrows the list of potential readers. Perhaps it would be possible to use some generalizations in the abstract?
  2. Also, the task in the abstract The distillation system under study was tested during July 2020 and 29 July was chosen as a typical day from 08:00 am to 8:00 pm, which is too specific for me. It might be worth mentioning that this period was.... more generally.
  3. I think a diagram explaining the description from line 133 to line 143 would be very helpful.
  4. The job description should be based on a diagram drawing beforehand, and then pictures of the various components.
  5. Why was this particular day chosen?
  6. In Figure 7 a strange approximation function is used, does this have any physical justification?

Author Response

(The authors gave the same response as above.)

Round 2

Reviewer 1 Report

The revised version of the manuscript "Experimental study of a tilt single slope solar still integrated with aluminum condensate plate" provides an interesting study about integrating an aluminum plate into a solar-driven salt-water distillation setup to improve distillation rates and efficiency. Overall, the methods and experiments are explained well and the manuscript is well organized. There is still room to further improve the manuscript. The discussions provided by the authors can be written more clearly. Some points for further improvement are written below:

  • [10] “productivity of conventional solar still” should be “productivity of a conventional solar still. There are still many grammatical errors throughout the manuscript that should be corrected prior to publication.
  • [148] – Explain what the angle of inclination is reduced from. (e.g. what is the previous system being considered and what was the angle in this initial case?).   
  • [164] – It should be clearly stated what is meant by “all other design conditions were assumed to be ideal”.
  • [219-220] – The reason for the glass cover requiring a longer time to raise its temperature is incorrect. The authors should rethink their interpretation and replace this discussion in the manuscript with a clear and correct description.

Author Response

Author's uploaded document

Reviewer 2 Report

The authors answered all the questions raised in my first report with sincerity and I greatly appreciated this. I confirm that the work is interesting and deserves to be published. As also agreed by the authors themselves, the work is still in its preliminary stages and several technological devices should be incorporated and included in the project (e.g., the solar distillation system in the current study is not equipped with a solar tracking system etc.). However, I believe that this study should be encouraged and I propose that the manuscript be published in its current version.

Author Response

Autors attached document file

Reviewer 3 Report

Dear authors,
thank you for your insightful replies and the changes you have made. I think that in its current version the article can be subjected to further publication procedures.
Yours sincerely
Reviewer

Author Response

Authors upoladed in document form
